

# A comprehensive exploration of machine learning techniques for EEG-based anxiety detection

Mashael Aldayel[1] and Abeer Al-Nafjan[2]

[1] Information Technology Department, College of Computer and Information Sciences, King Saud University, Riyadh, Saudi Arabia
[2] Computer Science Department, College of Computer and Information Sciences, Imam Mohammad Ibn Saud Islamic University (IMSIU), Riyadh, Saudi Arabia

## ABSTRACT

The performance of electroencephalogram (EEG)-based systems depends on the proper choice of feature extraction and machine learning algorithms. This study highlights the significance of selecting appropriate feature extraction and machine learning algorithms for EEG-based anxiety detection. We explored different annotation/labeling, feature extraction, and classification algorithms. Two measurements, the Hamilton anxiety rating scale (HAM-A) and self-assessment Manikin (SAM), were used to label anxiety states. For EEG feature extraction, we employed the discrete wavelet transform (DWT) and power spectral density (PSD). To improve the accuracy of anxiety detection, we compared ensemble learning methods such as random forest (RF), AdaBoost bagging, and gradient bagging with conventional classification algorithms including linear discriminant analysis (LDA), support vector machine (SVM), and k-nearest neighbor (KNN) classifiers. We also evaluated the performance of the classifiers using different labeling (SAM and HAM-A) and feature extraction algorithms (PSD and DWT). Our findings demonstrated that HAM-A labeling and DWT-based features consistently yielded superior results across all classifiers. Specifically, the RF classifier achieved the highest accuracy of 87.5%, followed by the Ada boost bagging classifier with an accuracy of 79%. The RF classifier outperformed other classifiers in terms of accuracy, precision, and recall.

# INTRODUCTION

Anxiety is a common mental health condition that can significantly impact individuals' abilities, behavior, productivity, and quality of life. It can manifest in various forms, such as generalized anxiety disorder, social anxiety disorder, panic disorder, and specific phobias, and can lead to severe emotional distress, physical symptoms, and impaired daily functioning. The impact of anxiety on individuals can be significant, leading to social withdrawal, avoidance of certain situations or activities, and interference with personal relationships. Anxiety can also have a significant economic burden on society, with a high cost of treatment and lost productivity (*Baghdadi et al., 2019*; *Mughal et al., 2020*).

Corresponding author
Mashael Aldayel,
maldayel@ksu.edu.sa

Early detection and management of anxiety can help reduce the economic burden associated with anxiety disorders and improve the overall well-being of individuals and society. Traditional diagnostic methods for anxiety disorders involve self-reported symptoms, which can be subjective and unreliable. Automated anxiety detection is required for anxiety management as it can help identify individuals who require early interventions and support. Automated anxiety detection using physiological signals provides an objective, and reliable method for detecting changes in mental states, which can be used in various settings to detect anxiety (*Nath & Thapliyal, 2021*; *Muhammad & Al-Ahmadi, 2022*).

Over the past few years, numerous studies have proposed automated anxiety detection using physiological signals, particularly electroencephalogram (EEG), galvanic skin response (GSR), and heart rate variability. These signals provide a reliable and non-invasive method for detecting changes in mental states, which can be used in various settings to detect anxiety. Accordingly, different artificial intelligence and machine learning approaches have been applied to build systems that have the potential to improve the accuracy of anxiety detection, thereby enabling early intervention and support for individuals with anxiety disorders (*Chatterjee, Gavas & Saha, 2023*; *Mane & Shinde, 2022*; *Chen et al., 2021*).

Anxiety disorder is difficult to cure, and its detection can be challenging due to ethical reasons. Researchers have proposed solutions to investigate these challenges, including the use of technology, algorithms, and signal processing techniques. A study by *Bubel et al. (2016)* aimed to validate the effectiveness of haptic feedback in determining rising anxiety levels and tested the device in a real-life public speaking situation to improve the device's design and functionality. Researchers have also proposed developing a compact, user-friendly, and precise universal emotion recognition system that utilizes sensors to detect emotions and produce valid results.

The challenges in acquiring physiological signals for anxiety detection include noise, baseline drifts, different artifacts due to body movements, different responses of participants to different stimuli, and low-graded signals. It is essential to have well-designed laboratories and carefully selected stimuli. Additionally, subjects must receive proper training to minimize variance in the data, and models must effectively generalize to new datasets or unseen data (*Mughal et al., 2020*).

This study aims to detect anxiety through EEG signals which are non-invasive, safe, and easy to measure. EEG signals provide valuable information about the brain's electrical activity, which can be used to identify changes in mental state. Thus, the study investigates the following research question: What is the most effective feature extraction and classification algorithm for detecting anxiety using EEG signals, and how to compare these algorithms to other approaches that have been used on the same dataset?

To achieve this, we proposed an ensemble learning approach and compare it with multiple machine learning models. Our proposed approach involved preprocessing the EEG signals, extracting features, and developing the classifier models. We evaluated the performance of the ensemble model using various metrics. The proposed ensemble

learning approach has the potential to improve the accuracy of anxiety detection using EEG signals and could be applied in various settings to detect changes in mental states.

The contributions of this study are twofold. First, we utilized a benchmark dataset to detect anxiety, which includes EEG signals recorded from individuals with anxiety disorders. Second, we conduct a comparative analysis of different feature extraction and machine learning algorithms and compare our results with other studies using the same dataset but different approaches. This comparison provides valuable insights into the strengths and limitations of different approaches and can guide future research in developing more accurate and reliable methods for detecting anxiety using physiological signals.

This article is organized as follows: "Background" presents the background. "Related Works" illustrates the literature review; "Materials and Methods" explains the experiment method; "Results and Discussion" discusses the results and the comparison of classical classifiers with related studies; and, "Conclusion" presents the conclusion and future works.

## BACKGROUND

### Anxiety using EEG signals

A brain-computer interface (BCI) is a computer-based system that acquires brain signals, analyzes them, and translates them into commands that are relayed to an output device to carry out the desired action. Thus, BCIs do not use the normal brain output pathways of peripheral nerves and muscles (*Aldayel, Ykhlef & Al-Nafjan, 2021*).

EEG is a technique used to record electrical activity in the brain. A passive EEG-based BCI approach for anxiety detection enables the extraction of valuable information from EEG signals without requiring active user engagement or explicit commands. By incorporating passive BCI techniques, there are open possibilities for real-time anxiety detection and monitoring in various applications. For instance, it could be integrated into wearable devices or mobile applications to provide individuals with personalized feedback and interventions to manage their anxiety levels. Moreover, the incorporation of passive BCI techniques in anxiety detection may contribute to the development of assistive technologies for individuals with anxiety disorders, enabling timely interventions and support.

Anxiety disorders involve extremely high levels of fear or concern that can alter the chemical properties of the brain. The amygdala, a small almond-shaped structure in the brain, plays a crucial role in processing emotions, including fear and anxiety. Individuals with anxiety disorders may have an overactive amygdala, leading to heightened fear and anxiety responses (*Mughal et al., 2020*).

EEG can provide rich information about central nervous system activities and linking mental states to brain activity. Anxiety has been found to be associated with activity in the amygdala, temporal, and prefrontal cortices, with the amygdala playing a key role in threat conditioning and response, valence, and salience (*Spampinato et al., 2009*).

Brain waves, measured in patterns of electrical activity in the brain, can be classified into five widely recognized frequencies: delta (0.5–4 Hz), theta (4–8 Hz), alpha (8–12 Hz), beta

(12–30 Hz), and gamma (above 30 Hz). EEG measures are sensitive to cognitive states such as task engagement, attention, working modality, and perception of user/machine errors, as well as individual moods and mental states including anxiety, surprise, happiness, and frustration (*Szafir & Mutlu, 2012*).

For instance, *Bai et al. (2020)* proposed an EEG emotion recognition system to distinguish between positive and negative emotional states in learners using the pre-processed version of the SEED dataset. They employed the wavelet transform approach to decompose and extract frequency bands and calculate sample entropy from EEG signals. A recurrent neural network and long short-term memory were utilized for emotion classification, achieving a final accuracy rate of 90.12%.

On the other hand, a recent study in *Al-Nafjan & Aldayel (2022)* explored the potential of EEG signals to objectively measure students' attention and engagement during online classes. The study extracted power spectral density (PSD) features using a fast Fourier transform and calculated different attention indexes. Three different classification algorithms, including k-nearest neighbors (KNN), support vector machine (SVM), and random forest (RF), were evaluated. The findings revealed that the proposed RF approach achieved a higher accuracy rate of 96% compared to KNN and SVM. This results of this study suggest that EEG-based attention detection systems can provide teachers with objective measures of student engagement, facilitating necessary adjustments during online classes.

## Ensemble learning

Ensemble learning is a powerful machine learning method that incorporates multiple models to improve predictive accuracy and reduce overfitting. Recent years have seen an increase in its popularity due to its ability to produce more accurate and robust models (*Dong et al., 2020*).

In ensemble learning, multiple models are trained on different subsets of data, then the predictions are combined to form the final prediction. There are a number of ways in which the base models can be combined. The models may be of the same type (homogeneous ensemble) or different types (heterogeneous ensemble), and the combinations may be achieved by using different methods such as averaging, weighted averaging, or stacking (*Dietterich, 2002*).

An important advantage of ensemble learning is its ability to reduce overfitting. By combining multiple models, ensemble learning captures different aspects of the data and reduces the likelihood of a single model being overfitted to the training data, which is particularly useful when dealing with complex datasets exhibiting high variability. Moreover, the accuracy of predictions can also be improved through ensemble learning. By combining the predictions of multiple models, ensemble learning can effectively reduce the variance in predictions and produce a more accurate final prediction. It is especially useful when dealing with noisy or uncertain data (*Dong et al., 2020*).

In ensemble learning, several methods are available, such as bagging, boosting, and stacking. The process of bagging involves training multiple base models on different subsets of the training data and combining their predictions by averaging them. In the

process of boosting, multiple base models are trained sequentially, with each subsequent model focusing on the data points that were misclassified by the previous model. The stacking process involves training multiple base models and using their predictions as input to a higher-level model (*Dietterich, 2002*).

The use of ensemble learning has been successful in a variety of fields, including finance, healthcare, and natural language processing. Ensemble learning, however, can be computationally expensive and may require a large amount of training data to be effective (*Sagi & Rokach, 2018*).

## RELATED WORKS

Over the past few years, numerous studies have proposed automated anxiety detection using physiological signals such EEG, GSR, and heart rate variability. While anxiety is a specific type of emotional state, it is important to distinguish it from other emotions. Emotion detection typically aims to recognize a broader range of emotions, such as happiness, sadness, anger, and fear. On the other hand, anxiety detection focuses specifically on identifying anxiety-related states. By developing specialized algorithms and models for anxiety detection, some previous studies aimed to enhance the accuracy and specificity of detecting this particular mental state, which is crucial for early intervention and support for individuals with anxiety disorders. Compared to studies on emotion recognition using EEG signals, there has been relatively little research conducted on anxiety detection within this field (*Baghdadi et al., 2019*).

To gain a better understanding of our research problem and the potential for utilizing EEG-based anxiety detection, we conducted a thorough literature review. Our review examined published research to provide insights to practitioners and researchers using the same dataset. Notably, the DASPS dataset is a recent publication from 2019, and consequently, few studies have employed this dataset to evaluate their methods.

*Baghdadi et al. (2019, 2021)* proposed a benchmark dataset named DASPS for the detection of anxious states. Two classification problems were formulated: two-level anxiety detection (light and severe) and four-level anxiety detection (light, normal, moderate, and severe). Several features were extracted for classification, including Hjorth parameters, fractal dimension, band power, Hilbert-Huang spectrum (HHS), discrete wavelet transform (DWT), and quantitative statistical EEG features such as spectral entropy, amplitude, connectivity, and range. The results showed that a Stacked Sparse AutoEncoder achieved an accuracy of 83.50%, outperforming KNN (81.40%) and SVM (77.40%) in classifying binary anxiety levels. The proposed benchmark dataset and classification methods provide a valuable resource for further research and development in the field of anxiety detection.

*Mane & Shinde (2022)* utilized the DASPS dataset to estimate mental stress levels and investigate the effectiveness of neural network techniques in utilizing EEG signals for this purpose. The study addresses the challenges of the irregular shape of EEG signals and the requirement for fixed data shapes as input to convolutional neural networks. It also explores the image-based spectrogram processing approach as a solution to this issue and its potential to improve the convergence efficiency of network models by considering the

multiparametric dependency of brain state estimation. Mainly, they convert EEG signals into azimuthal projection-based 2D images and employ convolutional neural networks for stress detection. The study evaluates the performance of the neural network model with varying numbers of convolutional layers and reports an accuracy rate of 93%.

*Muhammad & Al-Ahmadi (2022)* developed an objective framework for assessing human anxiety by utilizing EEG signals from the DASPS dataset. The study performed a channel selection step using statistical analysis techniques to identify significantly different electrodes, ultimately determining that channels AF3, AF4, FC5, FC6, P7, and P8 were statistically significant among the 14 channels of the headset. The study then extracted frequency domain features, including mean power, rational asymmetry, and asymmetry index, from the selected EEG channels. The frequency band selection algorithm was applied to identify the appropriate EEG frequency bands, which were found to be the theta and beta bands in this study. The wrapper method was employed for feature selection from all the features of the selected frequency bands. Results showed that a random forest classifier with nine features achieved an accuracy of 94.90% for two-level anxiety classification, while a random forest classifier with ten features achieved an accuracy of 92.74% for four-level anxiety classification.

*Chatterjee, Gavas & Saha (2023)* used EEG signals for the detection of mental stress. In the features extraction step, they proposed spatio-temporal transition-based features. Instead of working with the direct features computed from the recorded signals, they relied on the spatio-temporal transition behavior of those features. The results showed that the proposed method was able to discriminate various levels of mental stress in an individual with a maximum classification accuracy of 83.8% for both binary and four class classifications.

*Syakiylla Sayed Daud, Sudirman & Wee Shing (2023)* aimed to enhance the classification performance of an EEG-DASPS dataset by balancing it using a safe-level synthetic minority oversampling technique. The researchers filtered the raw EEG signals using multiple filtration methods and extracted features from the data in the time, frequency, and time-frequency domains to be used for model classification. They then processed the features model with the most optimal classification performance using a sampling technique and safe-level SMOTE before classifying it using KNN, SVM, and decision tree. The proposed model achieved an accuracy of up to 89.5% and a precision of 89.7% for the dataset with the enhanced class distribution using KNN. The findings suggest that the proposed method with safe-level SMOTE is more effective than existing methods without safe-level SMOTE in recognizing anxiety states.

Based on our review of the related works on anxiety detection using EEG signals, we aim to contribute to this research area by expanding upon the existing knowledge and addressing some of the identified gaps. Our main aims are to:

- Utilizing the DASPS dataset to investigate anxiety detection using EEG signals.
- Conducting a comparative analysis of feature extraction techniques and classification algorithms for anxiety detection.

- Exploring the potential of different classification techniques in estimating mental stress levels using EEG signals.
- Building upon previous research to enhance the understanding and development of EEG-based anxiety detection methods.
- Providing insights and findings to improve the accuracy and effectiveness of anxiety detection using physiological signals.

## MATERIALS AND METHODS

The results of anxiety detection depend on algorithm choices for feature extraction and classification. In this study, we examined the likelihood that two levels of anxiety, namely "anxious" and "non-anxious", that could be identified using a valence-arousal emotional model and different approaches of feature extraction and classification algorithms. For further improving of the accuracy of anxiety detection, we utilized the best results achieved from previous research *Baghdadi et al. (2019, 2021)*. Research *i.e.*, (trial duration of 1-s and best two approaches in feature extraction (PSD and DWT)). The analysis was performed in a trial duration of 1-s which was found more accurate for anxiety classification (*Baghdadi et al., 2019, 2021*). Moreover, we used their finding to improve the result by investigating the effect of different subjective measurements: SAM and HAM-A -based labeling with different machine learning models.

We chose the valence-arousal emotional model based on an analysis of neural correlations of anxiety which have been explained in a number of research studies (*Baghdadi et al., 2019, 2021*; *Spampinato et al., 2009*). For EEG feature extraction, we used DWT and PSD. We used PSD and DWT features because they have been shown to be effective for detecting anxiety in previous studies, (*Baghdadi et al., 2019, 2021*). They compared feature extraction across three domains (frequency domain, time domain and a time-frequency domain) and found that best results were achieved with time-frequency technique (DWT) and frequency techniques (PSD). In addition to their effectiveness, PSD and DWT features are also computationally efficient and easy to extract. Then, the valence-arousal emotional features were calculated for each approach. We applied intelligent computational modeling in the form of signal preprocessing and classification algorithms as these approaches can effectively reflect the subjects' anxiety states.

We applied different classification algorithms such as AdaBoost bagging and gradient bagging, random forest (RF), linear discriminant analysis (LDA), support vector machine (SVM), and k-nearest neighbors (KNN). Moreover, we compared the efficiency of classifiers for each approach of feature extraction. We implemented our model in Python programming language using different packages such as Scikit-Learn, numpy, SciPy, MNE, matplotlib and Keras for machine learning and signal preprocessing.

This section illustrates our methodology and provides implementation details for EEG-based anxiety detection as shown in Fig. 1. We begin with an explanation of the benchmark dataset. Then, we describe the anxiety model, signal preprocessing, data annotation/labeling, feature extraction, and computation phases. Finally, we describe classification algorithms for anxiety detection.

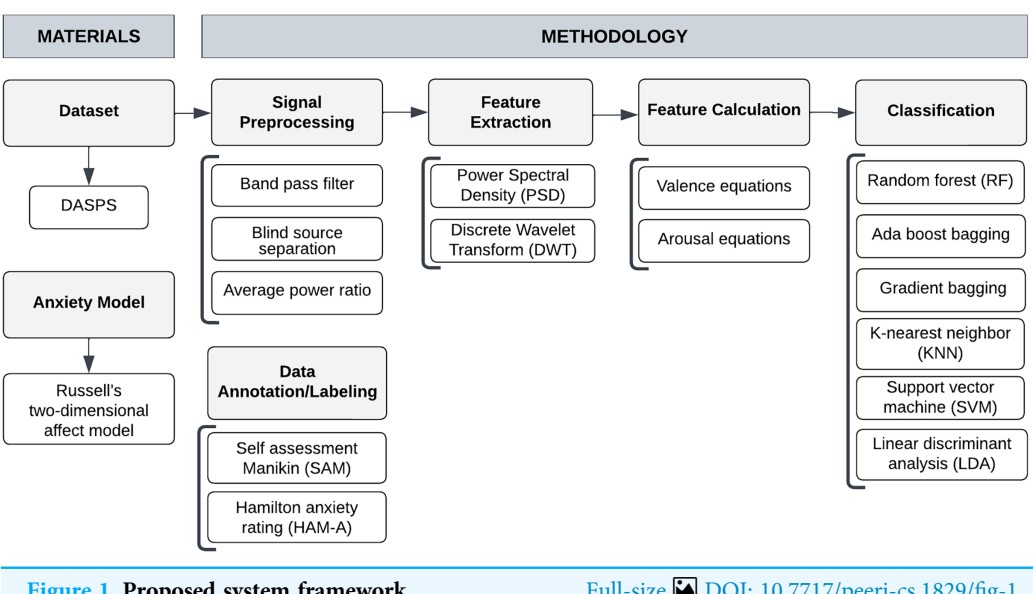

**Figure 1 Proposed system framework.**

| Table 1 Dataset description for anxiety detection. | |
|---|---|
| **Anxiety model** | **2-level (light and severe) and 4-level (normal, light, moderate and severe)** |
| Stimuli | Visual-based stimuli experiment of six situations *i.e.*, loss, family problems, financial problems, deadline, witnessing a deadly accident and mistreating). |
| Task | Flooding as *in-vivo* exposure therapy, *i.e.*, actual exposure to the feared stimulus that provoked the original trauma. |
| Subjects | 23 subjects (six trials for each subject) |
| Time | The recording includes six situations and took 6 min *i.e.*, (1 min per trial) |
| EEG device | 14-channel Emotiv EPOC |
| Experimental protocol | Psychotherapist conducts the HAM-A test for each participant and computes a total score that measures a subject's anxiety severity. For each situation, the psychotherapist recites each situation for 15 s then the subject recalls and imagines for another 15 s. Then the subject is asked to rate his/her feelings during stimulation using SAM for 30 s. After the 6-min trials, the psychotherapist re-evaluated some elements of the HAM-A test to update the subject's anxiety level |

## Dataset description

We used a public benchmark database named DASPS, "A Database for Anxious States based on a Psychological Stimulation", created by *Baghdadi et al. (2019)* for anxiety quantification into two and four levels.

Table 1 presents some aspects of the DASPS dataset. The original dataset includes EEG recordings gathered from 23 subjects (13 female and 10 male) with a mean age of 30 years old. These subjects were involved in exposure therapy that included confronting tolerable situations that trigger anxiety. DASPS dataset used flooding as *in-vivo* exposure therapy, which means actual exposure to the feared stimulus that provoked the original trauma. To identify the situations that caused the highest levels of anxiety, *Baghdadi et al. (2019)* surveyed all volunteers who wanted to participate in the experiment and found that the six situations that caused the highest levels of anxiety, in order of decreasing prevalence, were: loss (68%), family problems (64%), financial problems (54%), deadlines (46%), witnessing a deadly accident (45%) and being mistreated (40%).

## Anxiety model

For anxiety modeling, we used Russell's two-dimensional affect model to measure valence and arousal. Arousal is associated with intensity, whereas valence is associated with whether the emotion type is positive or negative. In DASPS dataset (*Baghdadi et al., 2019*), they quantify anxiety quantification into four levels- normal, light, moderate and severe-based on valance and arousal measures as shown in Fig. 2.

## Signal preprocessing

Our study aimed to detect a positive state of anxiety level. Therefore, we used only the 2-level labeling of anxiety states from the original dataset. We used the preprocessed version of the dataset that had been filtered with a band pass of 4–45 Hz. Artifacts had been removed with blind source separation, and components were selected based on the average power ratio of canonical correlation analysis. We further preprocessed the dataset to select the first second for each trial. We also chose 10 channels correlated to anxiety detection which are AF3, F7, F3, F4, F8, AF4, FC6, FC5, P7, and P8.

## Data annotation/labeling

Anxiety states were labeled using two different measurements: the Hamilton anxiety rating scale (HAM-A) and self-assessment Manikin (SAM). HAM-A measured the severity of subjects' anxiety at the end of the experiment. HAM-A is a 14-item self-report questionnaire that is used to assess the severity of anxiety symptoms. Each item is rated on a scale of 0 to 4, with a total score of 0–56. A score of 18 or higher is considered to be indicative of severe anxiety. SAM measured a two-dimensional emotion model based on valence (9-scale indicator varying from negative to positive) and arousal (9-scale indicator varying from sleepy to excited) for each trial in the experiment. Each scale is rated on a 9-point Likert scale, with higher scores indicating greater valence or arousal. HAM-A test was used to provide a quantitative measure of anxiety severity, and SAM test was used to provide a more qualitative assessment of the participants' current affective state. We used the HAM-A and SAM scores to label the EEG data as anxious or non-anxious.

For the SAM-based labeling, trials that had valence less than or equal to 5 and arousal greater than or equal to 5 were labeled as anxious states. The trials that had valence value ranging from 4 to 5 and arousal value ranging from 5 to 6 were labeled non-anxious as well as the remaining trials. Based on SAM-based labeling, there were 1,410 non-anxious and 2,730 anxious trials.

For labeling based on HAM-A that was measured at the end of the experiment, trials with HAM-A scores between 0 to 20 were labels signifying non-anxious while trials with HAM-A scores above 20 were labeled anxious. Based on HAM-A-based labeling, there were 1,440 non-anxious and 2,700 anxious trials.

Table 2 shows the number of instances of each anxiety level using both SAM and HAM-A based labeling. After labeling, we then used the labeled EEG data to train and evaluate our machine learning algorithms. To handle the imbalanced dataset, we used the Synthetic Minority Oversampling Technique (SMOTE) for both SAM and HAM-A labels. We chose to use SMOTE because it is a simple and effective oversampling technique that has been

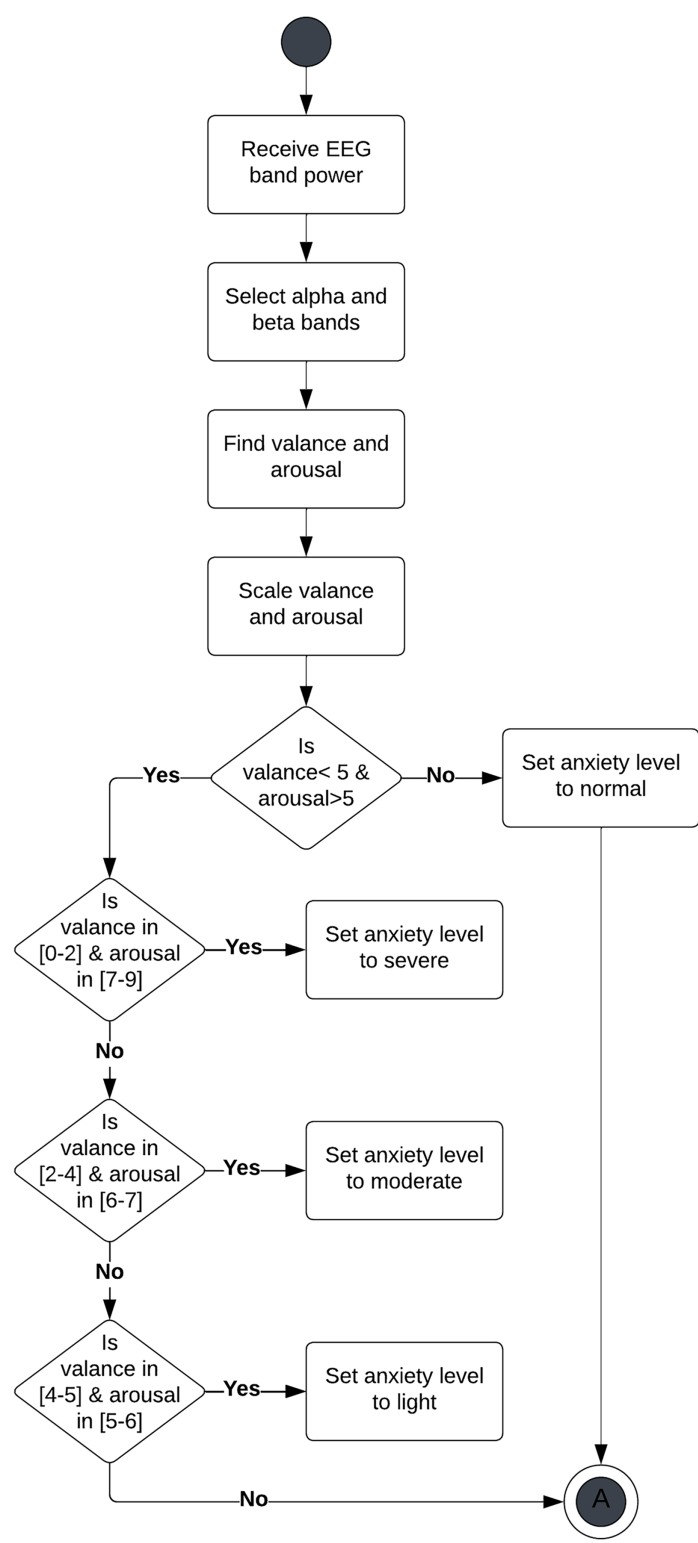

**Figure 2 Mapping anxiety into four levels based on valance and arousal measures.**

**Table 2 Anxiety labeling using SAM and HAM-A.**

| Anxiety label | SAM | HAM-A |
|---|---|---|
| Anxious | 2,730 | 2,700 |
| Non-anxious | 1,410 | 1,440 |

shown to improve the performance of machine learning algorithms on imbalanced datasets. However, it can potentially increase minority class clusters and lead to overfitting (*Nandini et al., 2023*). To address this concern, we compared the performance of machine learning algorithms with and without SMOTE. We found that SMOTE improved the performance of our algorithms on the held-out test set, without leading to overfitting. After oversampling in SAM labels, there were 2,700 non-anxious and 2,700 anxious trials. After oversampling in HAM-A labels, there were 2,730 non-anxious and 2,730 anxious trials.

## Feature extraction

The results of anxiety detection depend on algorithm choices for feature extraction and classification. In this study, we examined the likelihood that two levels of anxiety, namely "anxious" and "non-anxious", could be identified using a valence-arousal emotional model and different approaches to feature extraction and classification algorithms. We chose the valence-arousal emotional model based on an analysis of neural correlations of anxiety that has been explained in many research studies (*Muhammad & Al-Ahmadi, 2022*; *Baghdadi et al., 2021*).

For EEG feature extraction, we used DWT and PSD. Then, the valence-arousal emotional features were calculated for each approach. We applied intelligent computational modeling in the form of signal preprocessing and classification algorithms as these approaches can effectively reflect the subjects' anxiety states. Moreover, we compared the efficiency of classifiers, such as SVM, RF, and KNN for each approach of feature extraction. We implemented our model using Python programming language and different packages such as Scikit-Learn, numpy, SciPy, MNE and Keras for signal preprocessing, feature extraction and calculation and machine learning.

We extracted EEG frequency bands using two methods: DWT and a PSD. Then, we used the resulting frequency bands to calculate the valence-arousal affect features in each approach. The DWT method extracts a set of statistical features for each frequency band (details [D2–D5] and approximation [A5]). Whereas PSD stacks the extracted features into one array for each channel in the EEG signals. After applying PSD, we obtained 2023 features.

### Discrete wavelet transform

DWT is a multi-scale analysis based on frequency and time domains that breaks down signals into various and unique coefficients of brain signals. This decomposition results from the implementation of a set of high- and low-pass filters. The outcome coefficients of the low-pass filters are named approximation coefficients, whereas the outcome

**Table 3 Frequency bands correlated to decomposed coefficients.**

| Coefficient | Frequency (Hz) | Level |
| --- | --- | --- |
| D1 | 32–64 | $\gamma$ |
| D2 | 16–32 | $\beta$ |
| D3 | 8–16 | $\alpha$ |
| D4 | 4–8 | $\theta$ |
| A4 | 0–4 | $\delta$ |

coefficients of the high-pass filters are named detail coefficients. Based on the Nyquist theorem, the signal is down-sampled by a factor of two, resulting in a frequency band ranging between $f_n/2$ and $f_n$. The frequency of each detail coefficient is correlated to the sampling frequency rate $f_s$ of the raw signals, given by $f_n = f_s/2L + 1$ where $L$ is the decomposition level. Accurate DWT analysis depends on the value of $L$ and the choice of a suitable wavelet technique (*Vega-Escobar, Castro-Ospina & Duque-Munoz, 2016*; *Chen et al., 2015*; *Yadava et al., 2017*).

In this study, the $f_s$ was 128 Hz, $L$ was four levels and the wavelet technique was Daubechies (db4) wavelets. Hence, the decomposed EEG signals are five coefficients, called, D1, D2, D3, D4, and A4. Each one is related to the following frequency bands, respectively, (64–22 Hz) $\gamma$, (22–13 Hz) $\beta$, (13–8 Hz) $\alpha$,(8–4 Hz) $\theta$, and (4–1 Hz) $\delta$. Table 3 shows the decomposed coefficients (approximation A4 and details D1–D4) and their frequency bands.

In addition, we calculated the signal complexity using Shannon entropy which measures of the randomness or uncertainty in a signal. For a signal X, entropy is calculated as follows:

$$S(X) = -\sum_{i=1}^{N} p(x_i)\log_2(p(x_i)) \tag{1}$$

where p(X) is the probability of the signal value X. Moreover, we computed the statistical attributes that are frequently employed in signals, which are:

1. Mean: The average value of the signal.

2. Median: The middle value of the signal, when the signal is sorted in ascending order.

3. 25th and 75th percentile values: The values that divide the signal into four equal quarters, when the signal is sorted in ascending order.

4. Variance: The measure of how spread out the signal values are.

5. Standard deviation: The square root of the variance.

6. Root mean square of the average amplitude values: The square root of the average of the squared signal values.

7. Zero and mean crossing rates: The number of times the signal crosses the zero and mean values, respectively.

8. Mean of the signal derivatives: The average value of the derivatives of the signal.

We computed these ten statistical attributes for the entropy and each of the five coefficients in each of the 10 channels in the DWT-decomposed EEG signals. This resulted in a total of 600 features (10 attributes × 10 channels × (1 entropy + 5 coefficients)). DWT resulted in 608 features.

### Power spectral density

PSD measures the power of the signal based on frequency domain (*Xie & Oniga, 2020*). Previous research (*Chatterjee, Gavas & Saha, 2023*) has shown that the PSD generated from EEG signals is useful for detecting anxiety. With the PSD method, signals are converted from the time domain to the frequency domain and back again based on the fast discrete transformation of Fourier, and its inverse. In this study, we implemented the PSD method to decompose each signal into four frequency bands: $\gamma$ (40– 30 Hz), $\beta$ (30–13 Hz), $\alpha$ (13–8 Hz), and $\theta$ (8–4 Hz). The average power across the frequency ranges. In addition, we calculated the average and total power of each channel.

### Feature calculation

We used the following equations to calculate valence (2, 3, 4, 5) and arousal (6, 7, 8, 9). These equations were adapted from emotion detection research using EEG (*Al-Nafjan et al., 2017*)

$$\text{Valence 1} = \frac{\beta(AF3, F3)}{\alpha(AF3, F3)} - \frac{\beta(AF4, F4)}{\alpha(AF4, F4)} \tag{2}$$

$$\textit{Valence 2} = ln[\alpha(Fz, AF3, F3)] - ln[\alpha(Fz, AF4, F4)] \tag{3}$$

$$\textit{Valence 3} = \alpha(F4) - \beta(F3) \tag{4}$$

$$\textit{Valence 4} = \frac{\alpha(F4)}{\beta(F4)} - \frac{\alpha(F3)}{\beta(F3)} \tag{5}$$

$$\text{Arousal 1} = \frac{\alpha(AF3 + AF4 + F3 + F4)}{\beta(AF3 + AF4 + F3 + F4)} \tag{6}$$

$$\text{Arousal 2} = \frac{\beta(AF3 + AF4 + F3 + F4)}{\alpha(AF3 + AF4 + F3 + F4)} \tag{7}$$

$$\text{Arousal 3} = \log_2 \frac{\beta(Fz, AF4, F4, AF3, F3)}{\alpha(Fz, AF4, F4, AF3, F3)} \tag{8}$$

$$\textit{Arousal 4} = -(ln[\alpha(Fz, AF4, F4) + ln[\alpha(Fz, AF3, F3)]) \tag{9}$$

Using the Russell emotional model of valence and arousal, we mapped the outcome to investigate the possible anxiety level as explained in "Anxiety model".

### Classification algorithms

This study aims to detect two anxiety levels ("anxious" or "non-anxious") using EEG signals. Hence, intelligent machine learning algorithms were implemented to effectively reflect the anxiety of the users. Three ensemble classifiers RF, AdaBoost bagging and gradient bagging were proposed and their performance was compared to the KNN, LDA, and SVM classifiers. In addition, we compared the performance of the classifiers using different labeling (SAM and HAM-A) and feature extraction algorithms (PSD and DWT).

The hyperparameter selection process can be challenging, as there is no one-size-fits-all approach. The best way to select hyperparameters will vary depending on the machine learning algorithm, the data used, and the performance of the model. The hyperparameters for the machine learning algorithms were adjusted to the following:

- hyperparameters of KNN: The number of neighbors controls how many of the nearest neighbors are used to make a prediction. We tried different values of number of neighbors such as 1, 3, 5, 10 and adjusted it to 1 as it produces better results.
- hyperparameters of ensamble learning: The number of trees controls the complexity of the model. We tried different values of number of trees such as 100, 500 and 1,000.

  - AdaBoost bagging: The results shows that 100 is enough as no improvement achieved with 500 or 1,000. Therefore, we adjusted the number of trees in the forest to 100, which were all processed in parallel.
  - Gradient bagging: The results shows that 100 is enough as no improvement achieved with 500 or 1,000. Therefore, we adjusted the number of trees in the forest to 100, which were all processed in parallel.
  - RF: The results shows that 500 is good enough as no improvement achieved with 1,000. Therefore, we adjusted the number of trees in the forest to 500, which were all processed in parallel.

- hyperparameters of LDA: The regularization parameter controls the trade-off between model complexity and overfitting. We set the number of components to 1.
- hyperparameters of SVM: The kernel function controls how the similarity between data points is measured. We used the kernel of Radial Basis Function (RBF).

## RESULTS AND DISCUSSION

We detected the anxiety states of the subjects using two different feature extraction methods (PSD and DWT) and six classifiers: LDA, KNN, SVM, RF, AdaBoost bagging, and gradient bagging. For evaluation purposes, we split the data into 80% train and 20% test sets with holdout cross-validation and used various measurements for evaluating the classification algorithms such as precision, recall, and accuracy.

For PSD-based features, Tables 4 and 5 present the accuracy, recall, and precision results of the LDA, KNN, SVM, RF, AdaBoost bagging, and gradient bagging algorithms using anxiety labeling (SAM and HAM-A) with and without oversampling, respectively. All classifiers achieved better results after oversampling except SVM. SVMs can deal with unbalanced data because the class-weighted feature is assigning higher penalties for mis-classification in training objects of the minority class.

For DWT-based features, Tables 6 and 7 present the accuracy, recall, and precision results of the LDA, KNN, SVM, RF, Ada boost bagging, and gradient bagging algorithms using anxiety labeling (SAM and HAM-A) with and without oversampling, respectively.

Figures 3 and 4 analyze the classifiers precision from the viewpoint of anxiety labeling (SAM and HAM-A) for PSD and DWT feature extraction, respectively. The best results

**Table 4 Classifiers performance of PSD-based feature extraction with oversampling using SAM and HAM-A labeling.**

| | HAM-A label | | | SAM label | | |
|---|---|---|---|---|---|---|
| | Accuracy | Recall | Precision | Accuracy | Recall | Precision |
| KNN | 70.09% | 70.09% | 72.59% | 68.04% | 68.04% | 69.26% |
| LDA | 70.74% | 70.04% | 72.25% | 62.27% | 62.27% | 63.03% |
| SVM | 67.50% | 67.50% | 68.63% | 59.15% | 59.15% | 59.89% |
| RF | 84.07% | 84.07% | 84.15% | 79.40% | 79.40% | 79.50% |
| AdaBoost bagging | 76.11% | 76.11% | 76.29% | 69.32% | 69.32% | 69.41% |
| Gradient Bagging | 76.20% | 76.20% | 76.37% | 69.14% | 69.14% | 69.25% |

**Table 5 Classifiers performance of PSD-based feature extraction without oversampling using SAM and HAM-A labeling.**

| | HAM-A label | | | SAM Label | | |
|---|---|---|---|---|---|---|
| | Accuracy | Recall | Precision | Accuracy | Recall | Precision |
| KNN | 64.37% | 64.37% | 67.74% | 58.57% | 58.57% | 58.11% |
| LDA | 68.35% | 68.35% | 67.74% | 60.39% | 60.39% | 58.00% |
| SVM | 67.15% | 67.15% | 72.46% | 64.25% | 64.25% | 59.63% |
| RF | 78.74% | 78.74% | 78.67% | 68.36% | 68.36% | 67.47% |
| AdaBoost bagging | 74.52% | 74.52% | 74.20% | 67.03% | 67.03% | 65.60% |
| Gradient Bagging | 73.06% | 73.06% | 72.66% | 64.13% | 64.13% | 62.44% |

**Table 6 Classifiers performance of DWT-based feature extraction with oversampling using SAM and HAM-A labeling.**

| | HAM-A label | | | SAM Label | | |
|---|---|---|---|---|---|---|
| | Accuracy | Recall | Precision | Accuracy | Recall | Precision |
| KNN | 61.94% | 61.94% | 72.72% | 57.50% | 57.50% | 69.38% |
| LDA | 75.37% | 75.37% | 75.85% | 65.20% | 65.20% | 66.68% |
| SVM | 70.06% | 70.06% | 70.92% | 58.80% | 58.80% | 60.56% |
| RF | 87.50% | 87.50% | 87.65% | 82.14% | 82.14% | 82.32% |
| AdaBoost bagging | 78.98% | 78.98% | 79.03% | 74.43% | 74.43% | 72.54% |
| Gradient bagging | 77.87% | 77.87% | 77.92% | 73.44% | 73.44% | 73.52% |

were achieved with HAM-A labeling for all classifiers which can be justified as HAM-A rating scales were developed to measure the severity of anxiety symptoms, unlike SAM which measures emotional state based on valence and arousal.

As oversampling and HAM-A labeling provide better performance, the next paragraphs focus on investigating their results. Figure 5 analyzes the classifiers' precision from the viewpoint of feature extraction techniques (PSD and DWT). RF achieved the highest

**Table 7 Classifiers performance of DWT-based feature extraction without oversampling using SAM and HAM-A labeling.**

| | HAM-A label | | | SAM Label | | |
|---|---|---|---|---|---|---|
| | Accuracy | Recall | Precision | Accuracy | Recall | Precision |
| KNN | 64.00% | 64.61% | 65.11% | 55.56% | 55.56% | 55.89% |
| LDA | 72.58% | 72.58% | 72.03% | 60.62% | 60.62% | 59.08% |
| SVM | 70.04% | 70.04% | 72.27% | 64.25% | 64.25% | 41.28% |
| RF | 83.93% | 83.93% | 84.06% | 71.37% | 71.37% | 71.57% |
| AdaBoost bagging | 77.42% | 77.42% | 77.10% | 65.09% | 65.09% | 63.35% |
| Gradient bagging | 77.05% | 77.05% | 76.74% | 66.91% | 66.91% | 65.60% |

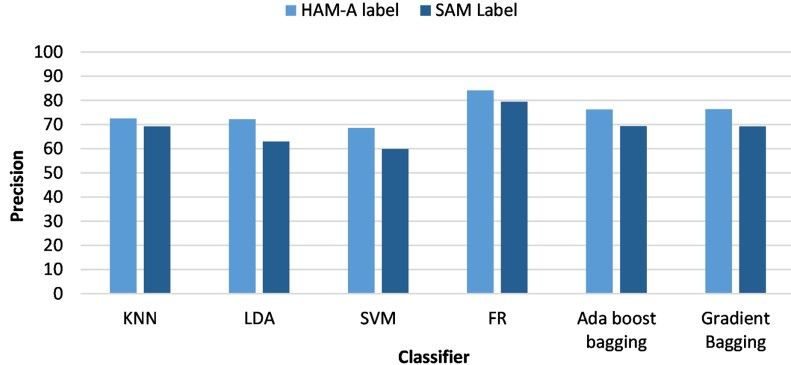

**Figure 3 Classifiers precision using different anxiety labeling (SAM and HAM-A) and PSD-based feature extraction.**

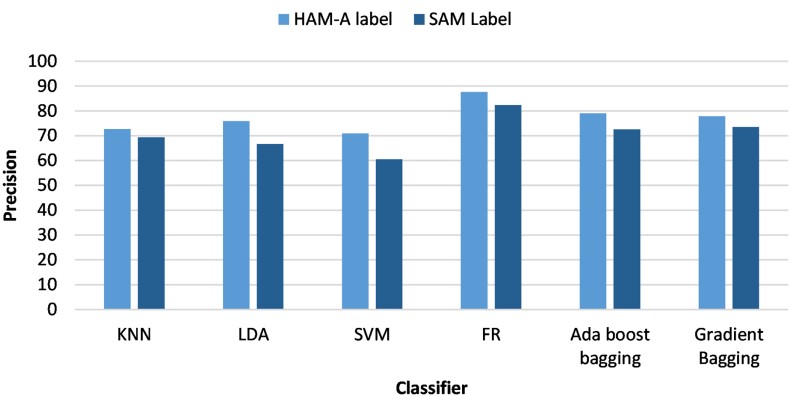

**Figure 4 Classifiers precision using different anxiety labeling (SAM and HAM-A) and DWT-based feature extraction.**

accuracy of all classifiers and DWT-based features achieved better results for all classifiers than PSD-based features. When using PSD-based features, the RF and bagging classifiers yielded enhanced accuracies of 84% and 76% with HAM-A labeling. Similar results were achieved with the DWT-based features. The highest accuracy was 87.5% with RF and the second-highest accuracy was 83% with AdaBoost bagging. Figure 6 gives the receiver

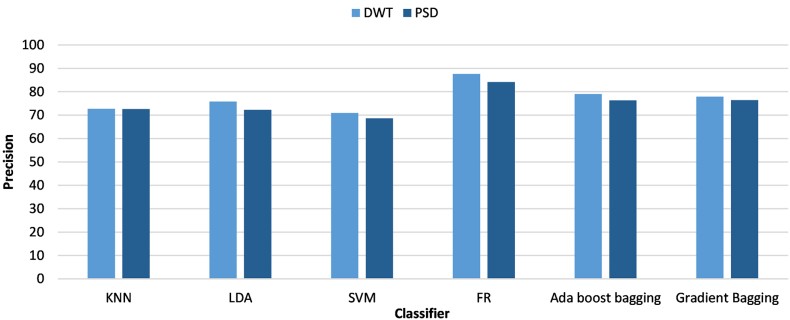

**Figure 5 Classifiers precision using HAM-A anxiety labeling and different feature extraction (PSD and DWT).**

operating characteristic (ROC) curves for four different experiments using the RF classifier. The experiments differ in the type of feature extraction used: PSD and SAM labeling, PSD and HAM labeling, DWT and SAM labeling, and DWT and HAM labeling. The area under the ROC curve (AUC) measures how well a classifier can distinguish between two classes, such as "anxious" and "non-anxious". A higher AUC value indicates a better classifier. The AUC values for the four experiments are: 0.7914, 0.8271, 0.8192, and 0.8781, respectively. This indicates that the best performing experiment is the one that uses DWT and HAM labeling for feature extraction. It is also worth noting that the false positive rate for all four experiments is relatively low, which indicates that the classifiers are good at avoiding false positives.

In addition to ROC, Fig. 6 presents the confusion matrix of RF to show how many instances of each class were correctly and incorrectly classified. The confusion matrix represents the class distribution of actual and predicted values for "anxious" and "non-anxious" classes.

We performed a statistical analysis that studies the concurrence (agreement) between two raters named Cohen's kappa coefficient. We used it to measure the agreement between the predicted labels resulted from RF and participants' subjective responses (HAM and SAM labeling). The kappa coefficient ranges from −1 to 1, where −1 indicates complete disagreement, 0 indicates agreement that is no better than chance, and 1 indicates perfect agreement. A kappa coefficient of 0.8 or higher is generally considered to indicate an almost perfect agreement between raters. The kappa values for the four experiments are: 0.599, 0.661, 0.628, and 0.757, respectively. The highest value is achieved by the experiment that uses DWT and HAM labeling for feature extraction which indicates that is the most effective feature extraction method for detecting anxiety using EEG signals.

In addition to the confusion matrix, ROC, and the statistical analysis results, the classification accuracy of our model was also compared to other previous studies that use similar approaches, where they used the same dataset but different extracted features and classification techniques as shown in Table 8.

As shown in Table 8, the combination of DWT-based feature extraction and RF in our study led to better results compared to features used in the other studies (such as Hjorth parameters, fractal dimension, band power, *etc.*). Moreover, we employed RF and

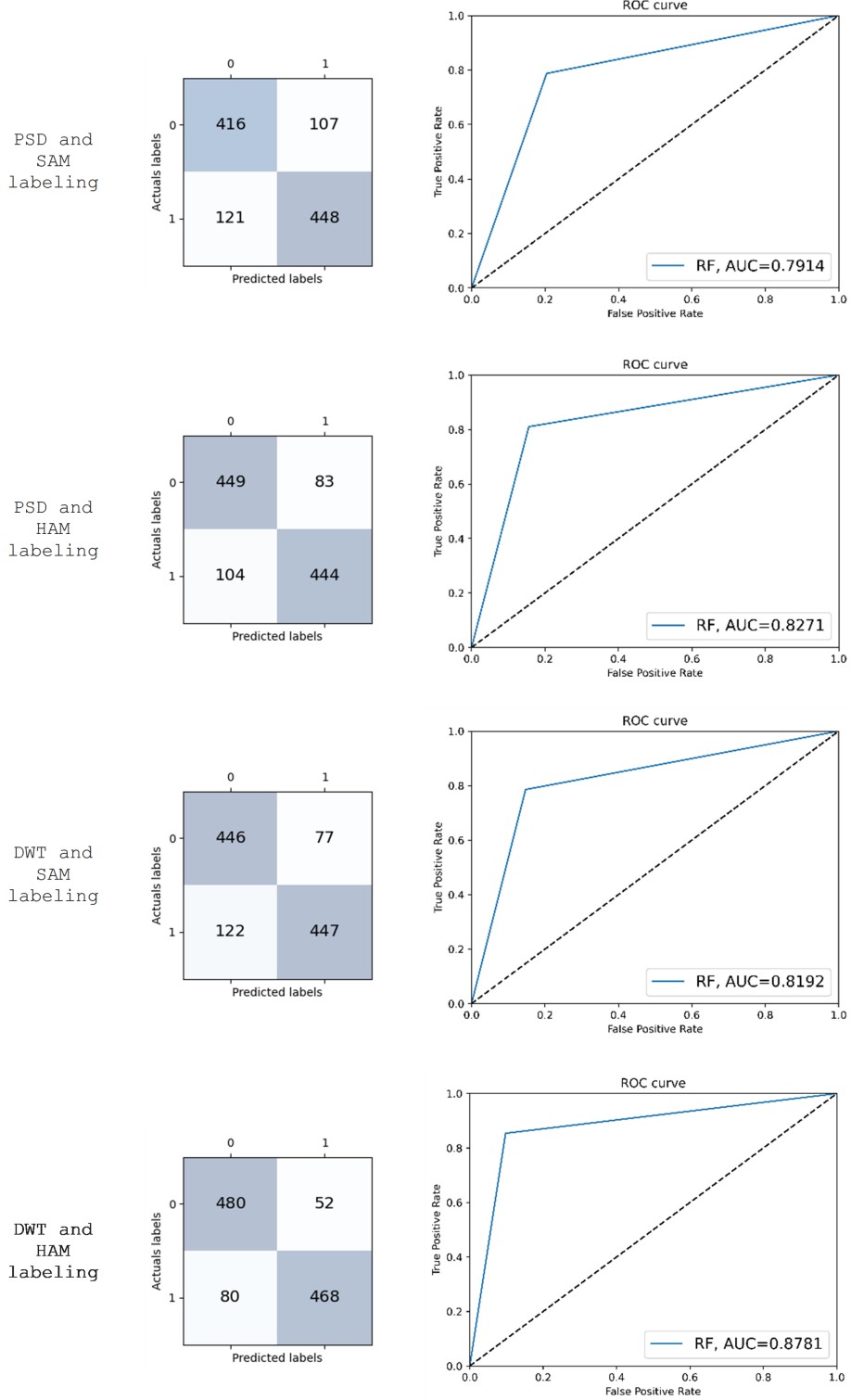

**Figure 6 Confusion matrix and ROC curve of the results obtained using RF.**

**Table 8 Related works for anxiety detection.**

| Ref. | Preprocessing | Feature extraction techniques | Class label | Classification algorithm | Accuracy |
|------|---------------|-------------------------------|-------------|--------------------------|----------|
| *Baghdadi et al. (2021)* | Pass-band filter, automatic artifact removal, Hamming-windowed Welch, blind source separation | Hjorth parameters, fractal dimension, band power, Hilbert-Huang spectrum, discrete wavelet transform, and quantitative statistical EEG features | Two-level anxiety | Stacked Sparse AutoEncoder | 83.50% |
| *Chatterjee, Gavas & Saha (2023)* | | Spatio-temporal transition-based features | Both two-level and four-level anxiety | KNN | 83.8% |
| *Syakiylla Sayed Daud, Sudirman & Wee Shing (2023)* | Bandpass filter, ICA, automatic artifact removal | Time-frequency domains sampling technique and safe-level SMOTE | Four-level anxiety decision tree | SVM | 86.0% |
| | | | | | 79.2% |
| Our approach | Band pass, channel selection, normalization | DWT and safe-level SMOTE | Two-level anxiety | RF | 87.5% |

AdaBoost bagging classifiers as none of the mentioned studies used them. Each study used different classifiers, such as stacked sparse auto-encoder, KNN, SVM, *etc*. The choice of classification algorithms can significantly impact the performance and accuracy of the anxiety detection model. In our study, the RF classifier achieved a higher accuracy of 87.5%, outperforming other classifiers, including the Ada boost bagging classifier with an accuracy of 79%. This indicates that the RF classifier was better suited for the feature combination in the same dataset.

## CONCLUSION

This study has investigated the use of EEG signals for anxiety detection by comparing different feature extraction and classification algorithms. We utilized a new dataset to detect anxiety and conducted a comparative analysis with other studies that used the same dataset with different approaches. Our results showed that the combination of HAM-A labeling and DWT-based feature extraction achieved better results across all classifiers, with the random forest and Ada boost bagging classifiers leading to enhanced accuracies of 87.5% and 79%, respectively. The performance of the RF outperformed other classifiers in accuracy, precision, and recall.

This study contributes to the growing body of literature on automated anxiety detection using physiological signals and provides a foundation for future research in this area. Our findings suggest that the use of EEG signals for anxiety detection has the potential to provide an objective, non-invasive, and reliable method for detecting changes in mental states, which can be used in various settings to detect anxiety. Although EEG-based anxiety detection is not yet ready for clinical use, but research is ongoing to improve its accuracy and reliability. Further research is needed to optimize the proposed approach and to investigate its effectiveness in real-world scenarios. The research limitations include the

complexity of EEG equipment and signal processing which can be include noisy signals mixed with meaningful signals. Moreover, the EEG signals vary widely from individual to another. This means that it is difficult to develop a single EEG-based anxiety detection algorithm that will work for everyone. As a part of our future work, we intend to experiment to record a comprehensive dataset and rigorously test our proposed model. This endeavor aims to enhance the accuracy and reliability of EEG-based anxiety detection, bringing it closer to clinical applicability.

### Funding

This work was supported and funded by the Deanship of Scientific Research at Imam Mohammad Ibn Saud Islamic University (IMSIU) grant number (IMSIU-RG23079). The funders had no role in study design, data collection and analysis, decision to publish, or preparation of the manuscript.

### Grant Disclosures

The following grant information was disclosed by the authors:
Deanship of Scientific Research (IMSIU): IMSIU-RG23079.

### Competing Interests

The authors declare that they have no competing interests.

### Author Contributions

- Mashael Aldayel conceived and designed the experiments, performed the experiments, analyzed the data, performed the computation work, prepared figures and/or tables, authored or reviewed drafts of the article, and approved the final draft.
- Abeer Al-Nafjan conceived and designed the experiments, performed the experiments, analyzed the data, prepared figures and/or tables, authored or reviewed drafts of the article, and approved the final draft.

### Data Availability

  The DASPS Data is available at Asma Baghdadi, Yassine Aribi, Rahma Fourati, Najla Halouani, Patrick Siarry, Adel M. Alimi, January 6, 2021, "DASPS database ", IEEE Dataport, doi: https://dx.doi.org/10.21227/barx-we60.
  The feature extraction functions and machine learning functions are available in the Supplemental Files.

### Supplemental Information

Supplemental information for this article can be found online at http://dx.doi.org/10.7717/peerj-cs.1829#supplemental-information.

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
