# Peer review of "A comprehensive exploration of machine learning techniques for EEG-based anxiety detection"

_PeerJ Computer Science, doi:10.7717/peerj-cs.1829_

## Round 0.1 · original submission · Major Revisions

I would appreciate it if the authors could address the issues raised by every reviewer. A double check of the whole paper would help as well.

**Language Note:** The Academic Editor has identified that the English language must be improved. PeerJ can provide language editing services - please contact us at copyediting@peerj.com for pricing (be sure to provide your manuscript number and title). Alternatively, you should make your own arrangements to improve the language quality and provide details in your response letter. – PeerJ Staff

Reviewer 1 ·

Basic reporting

1.There are lots of work for emotion recognition based on EEG, so what is difference between Anxiety detection and Emotion detection? The author made a summary about anxiety in Related Works, and I suggest the author make a brief discussion about the relationship of Emotion detection and Anxiety detection.
2. Figure 2-4 requires further modifications, such as horizontal and vertical coordinates, and so on.

Experimental design

The data used in this paper is from published DASPS dataset, it is ok for this study, but it is better the author run a experiment themselves.

Validity of the findings

The results seem good, but the comparison is just between different machine learning classifiers.
It is better to make a comparison with the benchmark methods, or the traditional method from these references.

Additional comments

This work based on a published anxiety dataset, proposed a method for anxiety detection. The main EEG feature is the PSD for each EEG-sub band, and combined with several classifiers. The results show the method is effective. Several suggestion should be considered :
1. it should give a discussion about the limitation of this study and the further work for anxiety detection in conclusion section.
2. the method proposed in this study should be compared with other methods from the reference.
3. the features used in this study is from Al-Nafjan (2017)'s work, which is used for emotional detection, is there any other better feature for anxiety detection. I think it is of course you can find other features, can you make a discussion, may in further work.
4. in Related Works, the authors make a summary of EEG based anxiety detection, so can the author discuss the relation or the difference for Anxiety and Emotion detection in the last part of this section.

·

Basic reporting

The manuscript discusses EEG signals- based anxiety detection using different combinations of feature extraction techniques, labelling approaches and ML algorithms. Prior to granting acceptance to this manuscript, the following aspects need careful consideration.

Basic Reporting
1. The title "Feature extraction and ensemble learning for EEG-based anxiety detection" is misleading since the authors employ established ensemble learning machine learning algorithms rather than introducing a novel one.
2. The abstract lacks the incorporation of keywords. Please include 4-6 keywords to enhance its comprehensiveness.
3. The English language should be improved to ensure that an international audience can clearly understand your text.
4. The reference citations within the text necessitate correction. Some erroneous and redundant reference citations are observable in lines 50, 170, and 190.
5. It is advisable to incorporate a systematic schematic diagram in Section 4, explaining the workflow of the proposed manuscript to enhance clarity and comprehension.
6. In Section 4, specifically lines 206-225, I recommend a revision to clearly outline the overall approach for the sake of clarity, as the current presentation is somewhat confusing.
7. The authors should incorporate the limitations of the present study and outline directions for future research in their manuscript.
8. The manuscript requires revision to adhere to a more structured format. The presentation of the proposed work is currently ambiguous and lacks clarity.
9. Incorporate a list of the main contributions as bullet points within the Related work section.
10. The sub-headings in the background section lack coherence. It could be integrated into the introduction section to provide a concise overview of the current landscape.
11. The manuscript does not address the potential applicability of the proposed research to a BCI system.
12. The caption is missing for the flow chart.
13. Conclusion section and Dataset description needs revision.
14. Please add labels to the x and y axes for Figures 2 to 4.

Experimental design

1. Please elaborate on the hyperparameter selection process in Section 4.7 and consider implementing hyperparameter tuning techniques for improved robustness.
2. Ambiguities in motivation, research gap identification, problem formulation, and articulating contributions persist in the manuscript. The novelty and advantages of the proposed research need to be emphasized by the authors.
3. Machine learning algorithms are typically evaluated using tools like the confusion matrix and ROC plot. I recommend their inclusion in both the Results and Discussions sections for a more thorough assessment of the work.
4. The comparison between a frequency domain and a time-frequency domain feature extraction technique may yield unfair results, as the latter has the capacity to investigate EEG signals in both domains. I recommend that the authors either conduct a comprehensive comparative study between time, frequency, and time-frequency domain techniques, or alternatively, focus their comparison solely on time-frequency techniques for a more equitable evaluation.
5. Comparing frequency and time-frequency domain techniques may yield biased results, as the latter examines EEG signals in both domains. Authors should either conduct a comprehensive comparative study of all three domains or focus on time-frequency techniques for fairness.
6. It is advisable for the authors to perform statistical analysis to validate their findings.
7. Integrate your findings into Table 1 and relocate the table to the Results and Discussions section. Additionally, provide a comprehensive analysis, comparing your results with the existing state-of-the-art methods available in the literature.
8. The authors should provide an explanation for their choice of using only the SMOTE method, which can potentially increase minority class clusters and lead to overfitting as discussed in DOI: https://doi.org/10.1016/j.bspc.2023.104894. Additionally, it is crucial to clarify whether they assessed and addressed the overfitting condition of their machine learning algorithms when employing this technique.
9. Please provide a clear motivation for the utilization of PSD and DWT, especially DWT, given that its use in anxiety detection has been previously discussed by Baghdadi et al. (2019, 2021), as mentioned in Section 3, Line 165 of the manuscript.
10. Provide a detailed explanation of the utilization of the HAM-A test and SAM test within the study.
11. The supplementary materials should be augmented with greater descriptive content, particularly by incorporating the source codes for the utilized machine learning algorithms.
12. Please elaborate on the feature extraction techniques mentioned in lines 299-304.

Validity of the findings

Please elaborate on the validity of the proposed study in real-life scenarios, considering the database used in the study is acquired in a controlled laboratory setting.

Additional comments

No comment

---

## Round 0.2 · Minor Revisions

There are still some issues to be addressed. A double check of the whole paper would help.

·

Basic reporting

I appreciate the authors for successfully incorporating the suggested changes in the paper. However, the following points still require revision:
1. The title "EEG-Based Anxiety Detection using Feature Extraction and Ensemble Learning" remains misleading, as the authors compare various standard machine learning algorithms and do not introduce a novel ensemble learning methods. The title should be revised. A more suitable option could be: "A Comprehensive Exploration of Machine Learning Techniques for EEG-Based Anxiety Detection" or "Exploring Anxiety Detection using Scalp EEG and Machine Learning Techniques."
2. The abstract should be restructured to emphasize the novelty of the presented work by following the problem, method, results, and conclusion structural order.
3. The Results and Discussions section is deficient in providing an in-depth analysis and discussion of the findings. Also, add dataset and signal processing columns to Table 8.
4. The Keywords section contains redundant information with the inclusion of both "Electroencephalogram" and its abbreviation "EEG."
5. The authors should incorporate information regarding data dimensionality after implementing all the feature extraction techniques.
6. Please incorporate the "%" symbol in Table 6-8 wherever it is necessary.

Experimental design

NA

Validity of the findings

NA

---

## Round 0.3 · accepted · Accept

Thanks very much for the revision.

·

Basic reporting

The current version of the paper can be accepted.

Experimental design

NA

Validity of the findings

NA